# De-Escalating Strategies in HPV-Associated Head and Neck Squamous Cell Carcinoma

**DOI:** 10.3390/v13091787

**Published:** 2021-09-08

**Authors:** Panagiota Economopoulou, Ioannis Kotsantis, Amanda Psyrri

**Affiliations:** Section of Medical Oncology, Department of Internal Medicine, Attikon University Hospital, National and Kapodistrian University of Athens, 1st Rimini St, Haidari, 12462 Athens, Greece; giotoik@med.uoa.gr (P.E.); ikotsantis@gmail.com (I.K.)

**Keywords:** HPV, de-escalation, deintensification, oropharyngeal cancer

## Abstract

HPV-related head and neck squamous cell carcinoma (HNSCC) has emerged as a diverse clinical and biological disease entity, mainly in young patients with oropharyngeal tumors who are nonsmokers and nondrinkers. Indeed, during the past few years, the pendulum has shifted towards a new epidemiological reality, the “HPV pandemic”, where the majority of oropharyngeal squamous cell carcinomas (OPSCCs) are attributed to HPV. The oncogenic potential of the virus is associated to its capacity of integrating oncogenes *E6* and *E7* into the host cell, leading to the inactivation of several tumor suppressor genes, such as Rb. HPV status can affect prognosis in OPSCC, but its role as a predictive biomarker remains to be elucidated. Given the favorable prognosis associated with HPV-positive disease, the concept of de-escalation treatment strategies has been developed with the primary intent being the reduction of treatment-related long-term toxicities. In this review, we aim to depict current data regarding treatment de-escalation in HPV-associated OPSCC and discuss ongoing clinical trials.

## 1. Introduction

Recent epidemiological evidence suggests a substantial rise of Human Papillomavirus (HPV)-related oropharyngeal squamous cell carcinoma (OPSCC), which accounts for more than 60% of OPSCC cases in the contemporary area [1,2,3]. Compared to tobacco-related disease, HPV-positive (HPV+) OPSCC is characterized by a distinct molecular landscape, including lower frequency of mutations in the *TP53*, *EGFR* and *CDNK1A* genes [4] and different demographic features, such as nonsignificant tobacco and alcohol consumption in a subset of cases and younger age at disease presentation, although an increased prevalence in older patients has been recently noted [5,6]. The combination of an early-stage primary tumor and bulky regional lymph nodes is a typical clinical presentation [7].

Most importantly, HPV+ OPSCC has been traditionally associated with significantly better outcomes [8,9,10,11]. In a landmark retrospective analysis of the Radiation Therapy Oncology Group (RTOG) 0129 trial in which patients with locally advanced head and neck squamous cell carcinoma (HNSCC) were randomized to radiotherapy (RT) with either accelerated fractionation with concomitant boost or standard fractionation with concurrent cisplatin, Ang et al. correlated HPV status with clinical outcome based on assessment of HPV DNA using in situ hybridization (ISH) [12]. It was postulated that patients with HPV+ OPSCC (63.8% of OPSCC cases who participated in the study) had a 58% decrease in the risk of death after adjusting for clinical and demographic features. Furthermore, Ang et al. suggested a patient risk stratification based on smoking status and TNM stage that defined patients with low- and intermediate-risk HPV+ OPSCC. Based on that classification, patients with intermediate-risk HPV+ disease are defined by a more than 10 pack/year smoking history and N2b/N3 or T4 disease by 7th edition staging; patients with HPV-OPSCC who are nonsmokers or have T1–T3 N0–N2 disease by TNM 7th edition belong to the low-risk group [12].

Given the remarkably better survival rates observed in HPV+ OPSCC, juxtaposed with the substantial morbidity caused by the standard of care treatment of cisplatin and RT combination in locally advanced disease, the concept of treatment de-escalation has been proposed as a new treatment paradigm. In this review, we seek to summarize current data on deintensification approaches in HPV+ OPSCC and discuss the goals of future de-escalation studies in this regard.

## 2. History and Rationale for Treatment De-Escalation-Selection of Patients 

The retrospective analysis of the RTOG 0129 trial published by Ang et al. identified a low-risk group of HPV+ OPSCC patients who exhibit high survival rates (90–95% at 3 years), implying a high chance of cure [12]. These results denote that a steadily increasing group of patients will survive several decades, albeit with significant treatment sequelae and functional abnormalities related to cisplatin-based chemoradiotherapy (CRT). Indeed, a pooled analysis of RTOG trials that used conventional RT techniques showed that 43% of treated patients experienced a severe late toxicity [13]. Although the incorporation of intensity modulated radiation therapy (IMRT) in the treatment algorithm of HNSCC has reduced the burden of treatment-related xerostomia and weight loss [14], late toxicities of CRT including deficits in speech, shallowing and mobility, poor oral health, trismus, and dysphagia greatly influence functional capacity of the head and neck area and severely diminish quality of life in long-term survivors [8,15]. In this context and given the young age and nonsignificant comorbidity of patients with HPV+ OPSCC, a framework of deintensification strategies has been proposed and assessed in randomized clinical trials with the goal of decreasing treatment-associated sequalae, without compromising tumor control.

The choice of suitable candidates for treatment de-escalation trials has become a matter of paramount importance. Two groups of patients should probably be excluded from participation. The first group (a) consists of patients with a smoking history (more than 10 pack years) due to a substantially worse prognosis, with a 3-year OS of approximately 70% [16]. Of note, compelling evidence from RTOG 9003 and RTOG 0129 studies have demonstrated that each year of smoking increased risk of death by 2% between p16-positive patients [17]. The second group (b) consists of patients who display adverse pathological features following surgery, such as extracapsular extension and positive margins. Several trials have used risk classification by Ang as inclusion criteria [12]. Despite being prospectively validated by two other cohorts [18,19], this classification system fails to incorporate relevant prognostic factors, such as biological features, HPV type, age, comorbidities, and response to treatment.

Importantly, the recognition that the American Joint Committee on Cancer (AJCC) 7th edition staging system does not adequately reflect the prognosis of HPV+ OPSCC, given the paradoxically favorable survival associated with the advanced stage, led to the development of the modified 8th edition clinical stage paradigm for HPV+ disease that has been more recently adapted [20]. However, despite improved prognostication conferred by the 8th edition AJCC staging [21], it cannot be utilized for guidance regarding patient management until clinical trials have validated alternative treatments based on 8th edition staging. Thus, since treatment decisions are still based on 7th edition staging, the new staging system mainly provides information for patient prognosis.

Alternative methods for risk stratification that are currently under investigation include the incorporation of genomic signatures [22], expression of molecular markers [23], and radiologic biomarkers for evaluation of response to treatment [24]. In addition, better selection of patients included in clinical trials, with a more stringent definition of an HPV positive status, including the analysis of presence of HPV DNA/or HPV RNA together with p16+ status and limiting studies to patients with OPSCC only in the tonsillar or base of tongue sites, may result in more accurate information on treatment de-escalation [25,26].

## 3. De-Escalation Strategies

Although there are many ongoing de-escalation strategies, the most commonly assessed in clinical trials are reduction of RT dose, field, or schedule when given either as adjuvant or definitive therapy and omission, dose reduction, or replacement of concurrent chemotherapy. Some of these clinical trials are presented below and in Table 1.

### 3.1. Reducing Radiotherapy Dose/Schedule

The cumulative dose of delivered RT has been substantially correlated with long-term functional deficits, which are lessened with doses below 60 Gy [27]. The rationale for reducing the dose of RT in HPV+ tumors is based on accumulated preclinical data [28] and reported outcomes from clinical trials in HPV+ patients (Lassen, JCO 2009) that indicate a higher radiosensitivity of HPV+ compared to HPV- disease. Thus, RT at a reduced dose might save patients with HPV+ OPSCC from long-term sequelae of RT without compromising efficacy.

*Definitive chemoradiotherapy.* In the setting of definitive chemoradiation, the phase II NRG-HN002 (NCT02254278) trial evaluated the efficacy of a reduced-dose RT (60 Gy) in patients with low-risk disease based on stratification by Ang’s criteria, namely patients with 7th edition stage T1-2N1-N2bM0 or T3N0-N2bM0 OPSCC with a ≤10 pack-year history of tobacco consumption [29]. Patients were randomized to either IMRT over 6 weeks concurrently with weekly cisplatin 40 mg/m^2^ or to IMRT alone over 5 weeks. This study showed that despite the satisfactory efficacy of concurrent CRT arm (2-year PFS of 90.5%), the RT alone arm did not meet acceptability criteria (2-year PFS 87.6%). Of note, 2-year OS was similar between the two groups.

In the same context, Chera et al. reported the results of a phase II trial that evaluated the combination of reduced dose IMRT (60 Gy) with cisplatin at a lower dose (30 mg/m^2^ weekly) [30]. The primary study endpoint was the pathologic complete response rate based on biopsy of the primary site and dissection of pretreatment positive lymph node regions. Eligibility criteria included patients with stage T0-3N0-N2c (7th edition AJCC) HPV+ OSCC with ≤10 pack-years smoking history. Past smokers were also permitted if they had ceased more than 5 years ago. The study met its primary endpoint of complete pathological response rate (86%, similar to standard CRT), which was estimated based on biopsy at primary site and dissection of positive lymph nodes before the initiation of treatment. In addition, de-escalated CRT resulted in a 3-year tumor control and survival of 100% while providing an improved quality of life, inter alia a significant reduction in feeding tube placement. In an almost identical single arm study conducted by the same group of authors, where a PET-CT scan was used to evaluate the need for LN dissection, 2-year PFS and OS were 86% and 95%, respectively [31].

In a similar context, EVADER is a single arm phase II study that seeks to evaluate the effectiveness of reduced RT volume to selected lymph node regions in patients with early stage (T1-3 N0-1M0 AJCC 8th edition) HPV+ OPSCC [32]. On the other hand, LCCC 1612 is an ongoing clinical trial that is using smoking status and molecular-based stratification as selection tools for treatment de-escalation. More specifically, the efficacy of reduced dose CRT (60 Gy) is being evaluated in patients with HPV+ OPSCC and (a) a history of light smoking (<10 pack years) and (b) with a history of >10 pack years smoking and no p53 mutations. On the contrary, patients with a history of > 10 pack years and p53 mutations will receive a standard dose of 70 Gy cisplatin-based CRT (NCT03077243).

*Deintensification of definitive RT after induction chemotherapy.* Response to induction chemotherapy (IC) has been used as a biomarker for selection of appropriate candidates for deintensified RT in phase II trials. All trials demonstrate improved outcomes compared to standard controls, warranting the design of phase 3 trials.

The Quarterback trial was a phase II study that enrolled patients with HPV+ disease and a <20 pack year smoking history who received induction TPF. Following IC, clinical responders (20 out of 23 patients) were randomized to receive a standard dose (70 Gy) or reduced dose (54 Gy) RT with weekly carboplatin [33]. In a median follow-up of 56 months, PFS and OS were similar between the two groups. In the ongoing Quarterback 2b trial, a reduced dose RT schedule (56 Gy) is being evaluated after response to IC (NCT02945631).

E1308 was a larger phase II trial with 90 participants, all stage III-IV (AJCC 7th edition) HPV+ OPSCC patients who were assessed for response at primary site and lymph nodes after IC with cisplatin, paclitaxel, and cetuximab. Patients with complete response (CR) were treated with RT at a reduced dose of 54 Gy, whereas patients with less than CR received standard dose RT of 69.3 Gy; both RT schedules were given in combination with weekly cetuximab [34]. Importantly, this study showed promising results, with a 2-year 80% PFS and 94% OS in IC responders treated with reduced dose RT. Importantly, functional outcomes, such as nutrition and shallowing capacity, were substantially improved at patients receiving low dose RT.

On the other hand, the OPTIMA phase II trial included patients with low risk (defined as ≤T3, ≤N2b, ≤10 pack-year smoking history) and high risk (defined as T4 or ≥N2c or ≥10 pack-year smoking history) HPV+ OPSCC, who, depending on their response at IC with a carboplatin/nab-paclitaxel combination regimen, were treated with reduced dose RT (low-risk with good response) or CRT (low-risk with medium response or high-risk with good response) or standard dose CRT (patients with any other response) [35]. Two-year PFS and OS were favorable for both low (95% and 100% respectively) and high-risk patients (94% and 97% respectively) with decreased rate of mucositis and PEG-tube use. Interestingly, OPTIMA II (NCT03107182) was a phase II trial which evaluated the same IC in combination with nivolumab; subsequently, patients were randomized to three treatment arms based on risk stratification and response to IC [36]. Patients were included in the high risk group if they had any of the following: T4, N2c-N3 (AJCC 7th edition), >20 pack-year smoking history, and nonHPV16 subtype; the remaining were stratified as low risk. Low risk patients with ≥ 50% reduction in tumor size were treated with transoral robotic surgery (TORS) or reduced dose RT (50 Gy) (Arm A). High risk patients with ≥50% reduction in tumor size and low risk patients with <50% reduction in tumor size were treated with CRT 45–50 Gy (Arm B), and the remaining patients were treated with standard CRT 70–75 Gy (Arm C). Adjuvant nivolumab was also administered for 6 months. Preliminary results were presented at American Society for Clinical Oncology (ASCO) 2021. The primary endpoint of deep response rate (DDR) after nivolumab-IC combination regimen was 70.8%, and 2-year PFS and OS were excellent (PFS: 96.3%, 85.8%, and 100% for arms A, B, and C, respectively, and OS: 96%, 91.9%, and 100% for arms A, B, and C, respectively). In addition, induction immunochemotherapy yielded a high pCR rate for patients who underwent TORS (66.7%) [36].

Finally, the RAVD trial similarly sought to assess the efficacy of a de-escalated approach that incorporated a reduced RT volume areas based on response to IC in patients with locally advanced HNSCC [37]. Consequently, patients with ≥50 % CR based on RECIST criteria received combined chemotherapy with reduced RT planning target volume (targeting gross disease), whereas patients with lesser response were treated with combined CRT at a planning target volume including elective nodal basins. For patients with HPV+ OPSCC included in the analysis, 2-year PFS and OS were 93.1%/92.1%, respectively, for good responders and 74.0%/95.2%, respectively, for nonresponders; PFS did not differ significantly between the two groups.

*Adjuvant therapy.* Adjuvant CRT is currently indicated in patients whose tumor pathology includes positive surgical margins or extracapsular lymph node extension. On the other hand, adjuvant RT is considered in tumors with positive lymph nodes and deep invasion (for primary site of the oral cavity).

The ECOG-ACRIN 3311 trial included patients with resectable p16+ OPSCC and cT1-T2 stage III/IV 7th edition AJCC without matted nodes. After transoral resection (TORS), patients were stratified depending on their pathological features to low risk (clear margins, 0–1 positive lymph nodes (LNs), no extracapsular extension (ENE)) who had no additional treatment (Arm A), intermediate risk (clear/close margins, 2–4 positive LNs, ENE ≤ 1 mm) who were randomly assigned to either postoperative reduced dose RT 50 Gy (Arm B) or RT 60 Gy (Arm C), and high risk (positive margins, >5 positive LNs, ENE >1 mm) who were managed with adjuvant cisplatin-based CRT (Arm D) [38]. For intermediate-risk patients, reduced dose RT demonstrated promising results, with a 3-year PFS of 94.9%, which met the primary endpoint of the study. For low-risk patients, 3-year PFS was 96.9% without additional RT. Importantly, 56% of patients in arms B and C vs. 36% in arm D reported stable or improved functional outcomes (*p* = 0.011). Based on this study, TORS plus adjuvant reduced dose RT should be evaluated in a phase III trial in comparison to standard radical CRT in patients with HPV+ OPSCC.

MC1273 was a phase II trial that sought to assess the efficacy of a remarkably reduced RT dose administered as adjuvant treatment following surgical management of stage III-IV (AJCC 7th edition) HPV+ OPSCC [39]. Eligible patients were non/light smokers (≤10 pack years) and were grouped as intermediate or high risk based on pathology (intermediate-risk: ≥T3, ≥2 positive LNs, lymphovascular/perineural invasion, LN > 3 cm; high risk: ENE, all patients had negative surgical margins). Patients in the intermediate-risk group were treated with 30 Gy RT delivered twice a day for 2 weeks in combination with weekly docetaxel, whereas patients in the high-risk group received an additional boost to LNs with ENE. The two-year locoregional control rate was 96.2% and the 2-year PFS was 91.1% in the total population; these outcomes were comparable to historical controls using standard treatment.

On the other hand, the AVOID trial, a phase II trial with distinct design, focused on patients with LN involvement after surgical treatment with TORS [40]. Sixty patients with stage pT1-T2 N1-N3 HPV+ OPSCC without other pathological risk factors (lymphovascular/perineural invasion, involved margins) were included. Adjuvant treatment following TORS consisted of deintensified RT to areas at risk in the involved neck (60–64 Gy) and uninvolved neck (54 Gy), sparing the resected primary site. Results were promising, with a two-year local recurrence-free survival of 97.9% and OS of 100%. Toxicity was favorable, with no patient requiring a feeding tube during RT; however, two patients required a temporal feeding tube during follow up.

PATHOS is an ongoing trial which transitioned into phase III after successful completion of the phase II cohort. Patients with stage T1–T3 N0–N2b HPV+ OPSCC are stratified to pathological subgroups following TORS and LN dissection. Patients in the low-risk group receive no further treatment, patients in the intermediate-risk group are randomized to standard or reduced dose RT, and patients in the high-risk group are randomized to CRT or RT [41,42]. The primary endpoint is noninferiority for OS with coprimary endpoint shallowing function at 12 months.

### 3.2. Chemotherapy Replacing

Given the significant toxicity correlated with cisplatin-based CRT and the promising efficacy of cetuximab-based bioradiotherapy in patients with OPSCC demonstrated in the IMCL 9815 trial [43], the idea of substituting cisplatin with cetuximab in patients with HPV+ disease has emerged as a tempting approach. Two randomized phase III trials, which constitute the only completed phase III trials in HPV+ OPSCC, have failed to corroborate noninferiority of cetuximab as compared to cisplatin in combination with RT. In the De-ESCALaTE HPV trial, 334 patients with stage T3N0–T4N0, T1N1–T4N3 (AJCC 7th edition) HPV+ OPSCC and limited smoking history (low risk by Ang stratification) [12] were randomly allocated to either standard high dose cisplatin-based CRT or RT in combination with weekly cetuximab [44]. The primary endpoint was rate of severe adverse events at 24 months of treatment completion. Interestingly, not only was there not any significant difference in severe toxicity between treatment arms, but the experimental arm was also associated with less favorable OS and (97.5% vs. 89.4%, *p* = 0.001) and 2-year relapse rate (6% vs. 16.1%, *p* = 0.0007).

RTOG 1016 was a noninferiority phase III trial published in same year as De-ESCALAaTE-HPV, which randomized patients with HPV+ OPSCC to either two cycles of cisplatin in combination with RT or cetuximab plus RT [45]. It was a larger trial that included 849 patients as compared to 334 in De-ESCALaTE-HPV and was not focused on low-risk patients. Indeed, 29% of patients were intermediate-risk based on stratification by Ang [12], and 16% had advanced T or N stage. Interestingly, the cetuximab/RT arm did not meet noninferiority criteria for OS, which was the primary endpoint of the study, and the cisplatin/RT arm yielded superior OS (5-year OS 84.6% vs. 77.9% in the cetuximab arm), with no differences in toxicity. Follow-up was 5 years compared to 26 months in De-ESCALaTE-HPV.

In addition to these two trials, TROG 12.0, a phase III trial conducted in Australia was recently reported at ASCO 2021 [46]. In this trial, 189 patients with AJCC 7th edition stage III (excluding T3N1) and stage IV (except from distant metastases and T4 +/− N3 +/− N2b-c if >10 pack years smoking history) disease were randomized to either cetuximab/RT combination or weekly cisplatin 40 mg/m^2^ and RT; primary endpoint was symptom severity [46]. Similar to the aforementioned trials, symptom severity did not differ between the two treatment groups, and 3-year failure free survival was worse with cetuximab (80% in the cetuximab-arm vs. 93% in the cisplatin arm, *p* = 0.015).

## 4. Conclusions and Future Directions

As we discuss in detail a variety of heterogeneous deintensification trials in HPV-related OPSCC, we understand that it is currently not the proper time for a broad application of de-escalation therapy in HPV+ patients outside of clinical trials. However, any patient with low-risk disease and no/limited history of smoking should be considered a candidate for participation in ongoing trials. A very tempting approach is the incorporation of immunotherapy in de-escalation strategies, since HPV+ OPSCC has an immunogenic profile and has been found to produce high response rates with immunotherapy [47]. Two trials investigate the substitution of cisplatin with antiPDL1 antibody durvalumab combined with RT in locally advanced disease (NCT03410615, NCT03623646), whereas a phase II study is assessing the safety and efficacy of neoadjuvant ipilimumab followed by nivolumab and reduced dose RT (NCT03799445).

Nevertheless, in addition to including a more stringent definition of HPV positive status and accurate information of the OSCC subsite, the identification and evaluation of novel biomarkers is a most important step in de-escalation trials and patient selection. The diagnosis of an HPV+ tumor can be improved in future clinical trials by adding the analysis of HPV DNA or HPV RNA to p16 immunohistochemistry, since around 10–15% of all p16+ oropharyngeal cancer is not HPV DNA+ [25]. Clinical biomarkers, such as disease stage and smoking history, and dynamic biomarkers such as response to induction chemotherapy, have been already widely used in reported trials. Molecular markers could potentially replace clinical biomarkers such as tobacco consumption, which is not easily quantifiable, and quantification is subjected to personal bias. Additionally, it is likely that poor prognosis associated with smoking history is due to genetic changes caused by tobacco and not tobacco itself. Genomic biomarkers that have been shown to correlate with dismal prognosis, such as PI3KCA mutation, could be used to exclude patients from those trials [48]. Imaging biomarkers such as MRI and 18F-FMISO PET are currently under investigation in ongoing trials (NCT03323463, NCT03224000). Most importantly, dynamic molecular biomarkers such as plasma or saliva ctDNA that have been shown to predict tumor recurrence in prospective studies [49,50] must be investigated in clinical trial protocols. Collecting serum and tissue samples to identify the characteristics of responders and favorable prognosis patients is fundamental in the design of de-escalation trials in HPV+ OPSCC and may produce robust algorithms to improve survival and guide patient management. In this regard, the REACT study is using circulating tumor DNA as a biomarker for treatment deintensification in low-risk HPV+ OPSCC; patients with high ctDNA levels at week 4 of treatment will be treated with standard dose RT or CRT, whereas patients with low ctDNA levels will be treated with reduced dose and schedule RT or CRT; results of this trial are eagerly awaited.

Ongoing de-escalation clinical trials in HPV+ OPSCC are shown in Table 1.

## Figures and Tables

**Table 1 viruses-13-01787-t001:** Ongoing clinical trials of treatment deintensification in HPV+ OPSCC.

Table	N (pts)	Phase	Stage/Eligibility	Treatment	Primary Endpoint
NCT04502407/ IIT2019-20-Zumsteg-HPVOPC	36	II	T0-3N0-2 p16 + OPSCC or cancer of unknown primary (AJCC 8th edition)	TORS→ - High risk pts (positive margins, ECS, ≥5 LNs): RT 50 Gy in 25 fractions, cisplatin 40 mg/m^2^ d1, 8, 15, 22 and 29 - All other pts: RT 30 Gy in 15 fractions, cisplatin 40 mg/m^2^ d1, 8, 15	2-year PFS
NCT02072148/ SIRS	200	II	Stage I, II, III or early and intermediate stage IVa (T1N0-2B, T2N0-2B) p16 + or HPV+ OPSCC	TORS→ Based on pathological features: Low risk: observation Intermediate risk: RT 50 Gy High risk: CRT 50 or 56 Gy	DFS and locoregional control at 3 years (high risk) DFS and locoregional control at 5 years (low/intermediate risk)
NCT03210103/ ORATOR2	61	II	Stage c T1-T2, N0-2 (AJCC, 8th ed.) p16 + or HPV+ OPSCC	Radiation +/− chemotherapy vs. transoral surgery+ LN dissection +/− RT	OS at 2 years
NCT03215719	54	II	T1-T2, N1-N2b or T3, N1-N2b (AJCC 7th Edition) p16 + OPSCC	Interval scan at 4 weeks post CRT: ≤40% nodal shrinkage: standard dose CRT >40 % nodal shrinkage: reduced dose CRT	PFS at 2 years
NCT04444869/ ENID	28	II	T1-T3, N0-N2c (AJCC, 7th ed.) p16 + OPSCC	Cisplatin-based CRT and RT dose de-escalation to clinically and radiologically uninvolved LNs	Rate of PEG tube placement
NCT03323463	300	II	T1-2, N1-2c HPV+ OPSCC	RT 30 Gy in 3 weeks + chemotherapy (cisplatin or carbo/5FU)	Effectiveness
NCT04900623/ReACT	145	II	Stage I, II, III (AJCC, 8th ed.) p16+ HPV+ OPSCC	Based on ctDNA levels High risk: standard dose RT/CRT in 7–8 weeks Low risk: low dose RT/CRT in 5–6 weeks	PFS at 2 years
NCT03875716/ ADAPT	111	II	Stage cT1-T2 cN0-N1 (AJCC, 8th ed.) p16 + or HPV ISH/PCR + cancer of tonsil/base of tongue	Based on pathology following curative-intent surgery with anticipated negative margins Low risk: observation Intermediate risk: RT 46 Gy High risk: RT 60 Gy without chemo	DFS at 2 years
NCT03410615	180	II	T1-2 N1 (smoking ≥ 10 pack years), T3 N0-N1 (smoking ≥ 10 pack years), T1-3 N2 (any smoking hx) (AJCC 8th edition) p16 + OPSCC	Standard cisplatin-based CRT vs. Durvalumab+ RT and adjuvant durvalumab vs.	NCT03410615
NCT03618134	82	Ib/II	Stage cT0-3 cN0-2b p16 + OPSCC	SBRT+ durvalumab → TORS + LN dissection (Cohort 1) SBRT+ durvalumab/tremelimumab → TORS+ LN dissection (Cohort 2)	Incidence of AEs, PFS at 2 years
NCT03799445	180	II	T1N2a-N2CM0, T2N1-N2CM0, T3N0-N2CM0 (AJCC 7th Edition) p16 + HPV+ OPSCC	Nivolumab + Ipilimumab + Reduced dose RT 50–66 Gy	DLT (safety lead in phase), CR, PFS
NCT03623646/ CITHARE	11	II	Newly diagnosed T1 N1-N2 or T2-T3 N0 to N2 (AJCC 8th edition) p16 + OPSCC	Radiotherapy + Cisplatin vs. Radiotherapy + Durvalumab	PFS at 12 months
NCT04638465	1000	observational	HPV+ OPSCC or unknown primary	Based on clinical stage: A (cT1-3 N0-1 tonsillar, c T1-2, N0-1 non tonsillar, cT0N1 unknown primary): TORS + LN dissection B (cT1-T3 N1 tonsillar, cT-T2 N1-N2): TORS +6 cycles of Cisplatin 40 mg/m^2^ C (cT1-T2 N2 tonsillar, cT0 N2 unknown primary): 6 Cycles of Cisplatin 40 mg/m^2^ + 60 Gy RT D (cT1-3 N3 cT4, any N tonsillar, cT3-4, any N, cAny T N3 non tonsillar, cT0N3 unknown primary): 7 Cycles of Cisplatin 40 mg/m^2^ + 70 Gy RT	OS and DFS at 10 years
NCT03601507	14	Ph II window	Clinical Stage I-IVA p16 + OPSCC	Neoadjuvant alpelisib → surgery	Quantitative change in the sum of RECIST measurable lesions, Change in tumor size in patients with genomic *PIK3CA* pathway alteration
NCT03342378	24	Observational	Stage III-IVB (AJCC 8th edition) intermediate or low risk HPV+ OPSCC	CRT (70 Gy with cisplatin 40 mg/m^2^) → PET/MRI prior the initiation of CRT, after 2 weeks of CRT and 3 months following the completion of CRT	Radiographic change in primary tumor and largest LN
NCT03952585	711	II/III	Clinical stage T1-2 N1M0 or T3 N0-1 M0 (AJCC 8th edition) p16 + OPSCC	IMRT+ cisplatin vs. reduced dose IMRT+ cisplatin vs. reduced dose IMRT + nivolumab	PFS, QOL
NCT03224000	75	II	Clinical stage T1-2 N0-2b M0 (AJCC 7th edition) p16 + or HPV DNA ISH OPSCC	MRI guided IMRT vs. standard IMRT	Locoregional control, Composite dysphagia outcome
NCT03077243/ LCCC 1612	215	II	T0-3 N0-2c HPV+ or p16 + OPSCC	Based on smoking history/p53 status: - ≤10 pack years → Reduced dose RT 60 Gy + cisplatin 30–40 mg/m^2^ weekly or cetuximab or Carbo/Paclitaxel or Carbo - 10 pack years, no p53 mutation → Reduced dose RT 60 Gy + cisplatin 30–40 mg/m^2^ weekly or cetuximab or Carbo/Paclitaxel or Carbo - >10 pack years, p53 mutation → Standard dose RT 70 Gy + cisplatin 30–40 mg/m^2^ weekly or cetuximab or Carbo/Paclitaxel or Carbo	2 year PFS
NCT02945631/ Quarterback 2b	65	II	Stage III-IV p16 + and HPV+ OPSCC	Reduced dose RT 56 Gy in 2 Gy fractions or 50.4 Gy in 1.8 fractions	PFS at 3 years

Abbreviations: AE = Adverse Event, AJCC = American Joint Committee of Cancer, CRT = ChemoRadiation, CR = Complete Response, DFS = Disease Free Survival, DLT = Dose Limiting Toxicities, HPV = Human Papilloma Virus, IMRT = Intensity Modulated Radiation Therapy, ISH = In Situ Hybridization, LN = Lymph Dissection, OPSCC = Oropharyngeal Squamous Cell Carcinoma, PEG = Percutaneous Endoscopic Gastrostomy, PFS = Progression Free Survival, TORS = TransoOral Surgery, QOL = Quality of life.

## Data Availability

Not applicable.

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
