# Peer review of "De-Escalating Strategies in HPV-Associated Head and Neck Squamous Cell Carcinoma"

_viruses, 2021, doi:10.3390/v13091787_

Round 1

Reviewer 1 Report

This review is a timely comprehensive review of present de-escalation trials that have been performed or are ongoing for treatment of human papillomavirus (HPV) positive oropharyngeal squamous cell carcinoma and provides useful knowledge for the scientific community.

However, some additional current information should be added in this context in order to bring the field further forward and provide improved information in future clinical trials.

  1. History and rational for de-escalation Selection of patients

In this chapter a historical perspective and an important rationale for treatment de-escalation is given. The important work by Ang et al [12] from 2010 is cited and the differences regarding staging of p16+ and p16- oropharyngeal cancer in the American Joint Committee on Cancer 7th and 8th edition are introduced. This important for the understanding of recent studies.

However, the authors also correctly point out that currently, the incorporation of genomic signatures, expression of molecular markers and radiologic biomarkers are under evaluation for their relevance and use for response to treatment.

Major points (but easy to address):

To be up to date, it is about time to add that the diagnosis of an HPV+ tumor can be improved in future clinical trials by adding the analysis of HPV DNA or HPV RNA to p16 immunohistochemistry, since around 10-15% of all p16+ oropharyngeal cancer is not HPV DNA+ and this is known very well by now. (Smeets SJ et al.  Int J Cancer. 2007 Dec 1;121(11):2465-72., Näsman et al. Papillomavirus Res. 2017 Dec;4:1-11. Furthermore, it is also known that HPV+ status (HPV DNA+, p16+ as a surrogate marker, or even more superior combined p16+ and HPV DNA+) is a very important prognostic factor for the two oropharyngeal subsites, tonsillar and base of tongue cancer subsites, but that the role of HPV-positive status in other oropharyngeal sites are not as evident for prognosis. (Marklund et al. Eur J Cancer. 2020 Nov;139:192-200). 

Thus in the future limiting de-escalated therapy to HPVDNA+, p16+ tonsillar and base of tongue cancer (and adding genomic incorporation of genomic signatures, etc.) may provide more accurate information in the future when conducting clinical trials with de-escalated therapy.  Therefore, to be up to date, in this chapter, and later in the concluding remarks the authors should, include information emphasizing that not all p16 tumors are HPV positive, and that most information on the positive impact of HPV+ status on prognosis is still only reflected with regard to the tonsillar and base of tongue oropharyngeal squamous cell carcinoma subsites.

Being more accurate in the selection of patients, including a better analysis and definition of HPV positive status and oropharyngeal subsite, and utilizing this information would therefore be useful in addition to genomic signatures and other biomarkers in future clinical trials.

Two suggestions to improve the manuscript are the following.

In the section 1, a last sentence could be added on line 94, and read e.g.

In addition, better selection of patients included in clinical trials, with a more stringent definition of an HPV positive status, including the analysis of presence of HPV DNA/ or HPV RNA together with p16+ status and limiting studies to patients with OSCC only in the tonsillar or base of tongue sites may result in more accurate information on treatment de-escalation [Smeets et al 2007, Marklund et al.2020].

In section 3, on line 286, the sentence could be improved, e.g.

Nevertheless, in addition to including a more stringent definition of HPV positive status, and accurate information of OSCC subsite, the identification and evaluation of novel biomarkers is a most important step in de-escalation trials and patient selection.  

Minor:

Line 24, two dots, one should be omitted.

Line 102. Add a final sentence here: Some of these clinical trials are presented below and in Table 1.

Line 134. Try rephrase to: In a similar context.

Table 1.

NCT03077243/LCCC 1612, for the last group >10 pack years, p53 mutations, I would assume it should read:  RT 70Gy

Author Response

We thank very much the reviewer for his/her kind words and comments and we agree with him/her. All changes are highlighted in yellow in the revised manuscript.

In the revised manuscript, we have added the two additional sentences in lines 94 and 286 and we have corrected all the minor points suggested by the reviewer (lines 24, 102, 134 and table 1)

Reviewer 2 Report

This is a well written and comprehensive review of de-escalating strategies in HPV-associated head and neck SCC. The rationale for de-escalation and de-escalation strategies are meticulously reviewed. De-escalation studies are well presented in table 1. However, what I lack is a short discussion about molecular markers in the conclusion, that potentially could replace clinical markers, e.g. smoking. My personal opinion is that smoking is a poor marker for patient selection, since it cannot be measured objectively and is hard to quantify. Besides, it is likely the genetic changes caused by smoking, and not smoking by itself, that causes the poor prognosis. Therefore, a molecular marker instead of an anamnestic marker would be more preferrable. I also lack a short sentence about HPV DNA/p16 instead of p16 only. Another minor point is that I would have preferred OPSCC instead of OSCC as an abbreviation for oropharyngeal squamous cell carcinoma.

Author Response

We thank the reviewer for her/his kind words and comments.

We have added a short discussion about molecular markers potentially replacing clinical markers in the conclusions session.

We have also added a short sentence about HPV DNA+/p16+ oropharyngeal cancer in the conclusion session.

Finally, we have also changed the abbreviation of oropharyngeal squamous cell carcinoma to OPSCC.